# Coordinated Auxin–Cytokinin–Nitrogen Signaling Orchestrates Root Suckering in Populus

**DOI:** 10.3390/ijms262412172

**Published:** 2025-12-18

**Authors:** Hongying Pang, Wanwan Lyu, Yajuan Chen, Liping Ding, Lin Zheng, Hongzhi Wang

**Affiliations:** 1Beijing Key Laboratory of Agricultural Genetic Resources and Biotechnology, Institute of Biotechnology, Beijing Academy of Agriculture and Forestry Sciences, Beijing 100097, China; phying123@163.com (H.P.); m15239623641@163.com (W.L.); chenyajuan@baafs.net.cn (Y.C.); dingluo2011@126.com (L.D.); 2Beijing Key Laboratory of Crop Molecular Design and Intelligent Breeding, Institute of Biotechnology, Beijing Academy of Agriculture and Forestry Sciences, Beijing 100097, China

**Keywords:** poplar root sucker, shoot apical meristem (SAM), hormone regulation, transcriptome analysis, Weighted Gene Co-Expression Network Analysis (WGCNA)

## Abstract

Root suckering is a key mode of clonal propagation in white poplar group, such as aspens (*Populus* section *Leuce*), enabling rapid vegetative spread, yet the molecular triggers remain elusive. Here, we developed a rapid protocol that produces abundant root suckers with the root cutting of white poplar (*Populus davidiana* × *P. bolleana*) roots in greenhouse. Anatomical analyses and daily resolution transcriptomes resolved three sequential developmental stages: primordium initiation (Days 0–1), SAM (shoot apical meristem) establishment (Days 1–4), and organ differentiation/growth (Days 4–6). Weighted gene co-expression network analysis revealed that auxin- and cytokinin-mediated signaling, integrated with nitrogen metabolism, orchestrates SAM formation and maintenance. Exogenous application of 0.5–1.0 mg L^−1^ NAA suppressed sucker emergence by 48–60%, whereas inhibition of cytokinin biosynthesis with lovastatin reduced initiation by 60%. These data establish that auxin negatively regulates and cytokinin is indispensable for de novo shoot apical meristem establishment during poplar root-suckering, underscoring that a precise auxin–cytokinin balance governs the timing and extent of this developmental process. Cambial regulators *WUSCHEL*-Related Homeobox 4-1/2 (*WOX4-1/2*), together with core meristem regulators *WUSCHEL* (*WUS*) and SHOOT MERISTEMLESS (STM), were specifically induced during SAM establishment that underpin vascular integration between the nascent shoot and the parental root. These results uncover the molecular pathway controlling root suckering and provide potential targets for molecular breeding to either enhance or suppress root suckering in *Populus*.

## 1. Introduction

*Populus* species pluralis (spp.) are the dominant fast-growing trees of temperate regions, yet most white poplars (*Populus* section *Leuce*) root poorly from cuttings. Consequently, grafting, tissue culture and root-suckering are the principal propagation routes [1,2,3,4]. Among these, root-suckering is especially advantageous for it is inexpensive, operationally simple, rejuvenates the clone and yields vigorous, rapidly established progeny [3,5,6]. Root suckers arise from adventitious buds on roots after canopy removal, mechanical wounding or root exposure [7] and are the primary mechanism by which North American aspen (*Populus tremuloides*) re-establishes entire stands following disturbance [7]. In China, commercial propagation of root suckers uses roots from 6- to 9-year-old trees [8]. Despite the clear influence of genetic background, endogenous hormone levels, nutrient availability, and environmental condition on root suckering [7,9,10], the underlying molecular mechanisms remains obscure. Elucidating it will inform both the promotion of desirable regeneration and the restricting of unwanted suckers, including transgenic escapes.

Auxin and cytokinin are the principal hormonal regulators of root suckering. Exogenous auxin application suppresses sucker emergence, and auxin-transport inhibitors such as α-(*p*-chlorophenoxy) isobutyric acid (CPIBA) markedly enhance it [11,12]. Conversely, exogenous cytokinin application promotes sucker initiation [13,14], and over-expressing cytokinin oxidase (CKX)-mediated depletion of endogenous cytokinin decreases it [15]. Decapitation, which lowers root auxin and simultaneously increases cytokinin concentrations, likewise stimulates suckering [7]. Parallel studies in the Rubiaceae show that a high cytokinin-to-auxin ratio in root favors root sprouting [16]. Meanwhile, de novo cytokinin biosynthesis and transport have been shown to be regulated by N signalling in plants [17]. However, how the auxin–cytokinin–nitrogen signals are dynamically orchestrated during root suckering remains unknown.

Primordia of root suckers originate from the cell of phellogen, where initial meristematic activity recruits parenchyma cells of the phelloderm, pericycle, and phloem to re-establish a functional shoot apical meristem (SAM) that subsequently drives sucker development [7,18,19]. A SAM comprises three cytohistological zones: the central zone (CZ), rib zone (RZ), and peripheral zone (PZ) [20,21] (Figure 1a). The CZ functions as a stem-cell niche characterized by relatively slow mitotic activity and replenishes the RZ and PZ through precisely controlled divisions, whereas PZ- and RZ-specific programs drive lateral organ initiation and axial growth, respectively. Transcriptional profiling has revealed distinct molecular signatures for each zone [20,21,22]. In particular, *WUSCHEL* (*WUS*) and *SHOOT MERISTEMLESS* (*STM*) have been identified as stem-cell identity genes and indispensable CZ regulators; loss-of-function mutants for either gene fail to establish a functional SAM [23,24,25,26]. Ectopic *WUS* expression alone is sufficient to induce shoot regeneration from roots or callus in the absence of exogenous hormones [27,28]. *WUS* expression is confined to a specific domain—the *WUS*-expression domain—within the CZ, which is essential for stem-cell and SAM homeostasis. A negative-feedback loop between *WUS* and the CLAVATA3 (CLV3)–CLV1/BAM1 pathway plays an essential role in maintaining *WUS* expression homeostasis [29,30,31].

Cytokinin and auxin have been shown to play essential roles in regulating the spatiotemporal expression of *WUS* during de novo SAM formation and maintenance. In Arabidopsis, key players in cytokinin signaling—the type-B Arabidopsis response regulators (ARR1, ARR2, ARR10, ARR12)—bind directly to the promoter of *WUS* and activate its transcription [27,32,33]. Accordingly, the triple *arr* mutant (*arr1 arr10 arr12*) exhibits almost undetectable *WUS* transcripts and fails to regenerate shoots from root explants. Conversely, auxin represses *WUS* transcription; auxin-overproducing mutants display reduced *WUS* levels and impaired shoot regeneration [34]. Indeed, within the *WUS*-expression domain, no auxin response signal is detected, whereas a strong cytokinin response signal is present [35]. This differential distribution—with high cytokinin in the CZ and high auxin in the PZ—has been shown to be established through reciprocal repression of hormone biosynthesis. For example, ARRs negatively regulate YUC1/4-dependent auxin biosynthesis [33], and the auxin response factor ARF3 represses IPT5-mediated cytokinin biosynthesis [35]. This reciprocal antagonism facilitates the distinct roles of cytokinin in stem-cell maintenance in the CZ and of auxin in organogenesis in the PZ, respectively. Although these mechanisms have been well characterized in the aerial SAM and/or during in vitro regeneration, how these pathways are reprogrammed to establish a SAM de novo on a root during root suckering remains unexplored.

Here we present an efficient, rapid protocol for inducing root suckers in greenhouse with *Populus davidiana* × *P. bolleana* (Shanxin yang), a widely planted hybrid in northern China. Using high-resolution transcriptome profiling, we dissect the temporal dynamics of root suckering, identify key regulatory pathways, and propose candidate genes. Our results reveal that coordinated auxin, cytokinin, and nitrogen signaling are central to the de novo formation and maintenance of a functional SAM during root suckering.

## 2. Results

### 2.1. Rapid Induction System and Temporal Transcriptome Dynamics of Root Suckering

To elucidate the molecular mechanisms that drive root suckering in poplar, we first established a rapid and highly efficient induction system using three-year-old white poplar (*Populus davidiana* × *P. bolleana*) roots collected after natural leaf fall in autumn (Figure 1b). When root segments were incubated in soil under controlled warm greenhouse conditions, suckering began almost immediately. White, ellipsoidal protuberances appeared by day (D) 3, progressed to easily discernible buds with small juvenile leaves by D5–6, and yielded vigorously growing shoots by D14 (Figure 1d). Semi-thin sections of 3-day-old protuberances revealed a phellem layer enveloping a root-borne shoot primordium (Figure 1c). Based on well-established anatomical criteria for meristematic cells, this primordium contained two procambial strands and an apical cluster of meristematic cells (Figure 1c). The latter are located at the distal tip of the nascent primordium, directly apical to the two procambial strands, and are destined to become the functional SAM of the adventitious shoot. These procambial strands connected the nascent meristem to the root vascular cambium, ensuring vascular continuity between the future shoot and the parental root.

Next, we performed daily sampling (D0–6) during root suckering for transcriptome sequencing. All reads were mapped to the *Populus trichocarpa* v4.1 reference genome (Phytozome: https://phytozome-next.jgi.doe.gov/info/Ptrichocarpa_v4_1, accessed on 13 May 2024). Pearson correlation coefficients ranging from 0.945 to 1.000 among biological replicates confirmed high reproducibility (Figure 2a). Principal-component analysis resolved four distinct transcriptional states corresponding to D0, D1, D2–4 and D5–6 (Figure 2b). These four transcriptional clusters infer three developmental stages of root suckering: (i) primordium initiation (D0–1), (ii) SAM establishment (D1–4), and (iii) organ differentiation and growth (D4–6). Across all libraries, 15,000 genes were detected as expressed (Figure 2c; Appendix A). Pairwise differential-expression analyses (D1–6 versus D 0) identified 14,305 differentially expressed genes (DEGs), 4913 of which were shared across all contrasts (Figure 2d). Each individual comparison yielded approximately 9000 DEGs (Figure 2e), reflecting continuous and dynamic transcriptional reprogramming throughout root suckering.

### 2.2. WGCNA Reveals Stage-Specific Key Pathway and Genes During Root Suckering

To systematically identify the gene sets most tightly associated with root-suckering progression, we subjected the 15,000 expressed genes to Weighted Gene Co-expression Network Analysis (WGCNA). Fourteen distinct modules were resolved (Appendix A), six of which showed significant correlations with specific developmental stages (*p* < 0.05, |r| > 0.44; Figure 3a,b). Specifically, the turquoise, green-yellow, and red modules were linked to primordium initiation (D0–D1); the brown and green-yellow modules to SAM establishment (D1–D4); and the blue, pink, green-yellow, and red modules to organ differentiation and growth (D4–D6) (Figure 3c). These six modules therefore represent the core transcriptional programs underlying root suckering and were selected for detailed downstream analysis.

Functional enrichment of the three D0–D1-linked modules revealed a rapid transition from dormancy to high metabolic activity in root segments. KEGG analysis (*p* < 0.05) highlighted ribosome, ribosome biogenesis, the tricarboxylic acid cycle (TCA cycle), oxidative phosphorylation, nucleocytoplasmic transport, aminoacyl-tRNA biosynthesis, and the biosynthetic or metabolic pathways for glycine/serine/threonine, histidine, and branched-chain amino acids (Val, Leu, Ile), indicating global activation of respiratory and protein-synthetic machinery upon transfer to warm, favourable growth conditions (Appendix A). GO terms further pinpointed a coordinated early response (Figure 4a and Appendix A): the tryptophan biosynthetic process (GO:0000162), containing *TSB1*, *TSB2* and *PAT1*, whose transcripts were dramatically up-regulated at early induction stage (Figure 4b and Appendix A) and may support local auxin synthesis by supplying the precursor [36]; the hormone-response category (GO:0009725), including several auxin-response factors, among which *ARF5* was markedly induced at the early stage and remained high throughout root suckering (Figure 4b,g); and regulation of the mitotic metaphase/anaphase transition (GO:0030071), featuring induction of *APC*/*C* genes *APC4*, *APC8* and *APC10* at the early stage (Figure 4a,b), whose products drive cyclin-B1 degradation to re-initiate cell division [37]. Collectively, the rapid up-regulation of auxin and cytokinin biosynthesis and signalling, coupled with cell-cycle re-entry, constitutes the earliest molecular commitment to the formation of sucker primordium in root.

GO enrichment of the D1–4-associated modules revealed a pronounced nitrogen signature, with nitrogen compound metabolic process, cellular nitrogen compound metabolic process, regulation of nitrogen compound metabolic process, and cellular nitrogen compound biosynthetic process all significantly over-represented (*p* = 5.24 × 10^−7^, 3.24 × 10^−6^, 4.14 × 10^−5^, and 8.72 × 10^−5^, respectively; Figure 4c and Appendix A). Within these categories, enzymes central to aspartate and glutamate metabolism—including ASP1/ASP2 (aspartate aminotransferases), CTP synthase, GAD (glutamate decarboxylase), and AK2 (aspartate kinase)—were sharply up-regulated during SAM establishment (Figure 4d,g), linking N assimilation and metabolism to this process. Concurrently, 19 hormone-response genes were enriched, 18 of which were auxin-responsive *ARFs* or *SAURs* (e.g., *SAUR78* and *SAUR32* peaking during SAM establishment) alongside activated cytokinin-response genes *RR9-1*, *RR9-2* and the cytokinin receptor *AHK4* (Figure 4d,g), indicating that auxin–cytokinin crosstalk continues to orchestrate SAM formation in the context of nitrogen metabolism.

KEGG analysis of the D4–6 modules identified “Photosynthesis” and “Photosynthesis—antenna proteins” as the two most enriched pathways (Appendix A), while GO analysis underscored photosynthesis, carbohydrate biosynthesis, leaf and phyllome development, and plant-type cell-wall organization (Figure 4e)—all hallmarks of active SAM-derived organogenesis. Nitrogen metabolism remained prominent, with glutamine-family amino-acid metabolism, glutamine metabolic process, and glutamine biosynthetic process significantly enriched; the cytosolic glutamine synthetase genes *GS1-1*, *GS1-2*, and *GLN2* peaked at this stage (Figure 4f and Appendix A). Concurrently, the enriched intracellular-signal-transduction category contained cytokinin signaling components—the receptor *AHK1* and type-B response regulators *RR4*, *RR5*, and *RR9*—which reached maximal expression at this stage (Figure 4f,g). These data indicate that sustained cytokinin signaling and nitrogen metabolism jointly underpin SAM maintenance and continued organogenesis.

After analyzing the functional enrichment of the stage-specific modules, we validated key transcriptomic trends by RT-qPCR. The expression patterns of representative genes involved in auxin and cytokinin signaling and nitrogen metabolism—including *ARF5*, *AHK4*, *APS1*, and *APS2*—were independently confirmed (Figure 4g), supporting the RNA-seq data and reinforcing the WGCNA-based developmental staging.

### 2.3. Molecular Pathways of Auxin and Cytokinin Governing SAM Formation in Poplar Root Suckering

The pivotal roles of auxin and cytokinin in the initiation and maintenance of aerial SAMs are well established [38,39]. In Arabidopsis, the two principal cytokinin types—trans-zeatin (tZ) and N^6^-(Δ^2^-isopentenyl) adenine (iP)—display distinct spatial distributions and functions, with tZ being indispensable for normal SAM establishment [40,41]. Each class is synthesized by a dedicated set of genes (Figure 5a). To dissect the molecular basis of root suckering, we first monitored the expression dynamics of cytokinin-related genes involved in synthesis, metabolism, and transport. These genes resolved into three clusters (Figure 5b). Cluster I comprises seven genes, most of which encode biosynthetic enzymes (*IPTs*, *LOGs*, and *CYP735As*) and were sharply up-regulated immediately after induction. Notably, *CYP735A2*—which channels the intermediate iP riboside 5′-phosphate (iPRP) into the tZ pool [42]—showed the strongest and most stage-specific induction during SAM establishment (D2–D4) (Figure 5a,b). Arabidopsis *cyp735a1 cyp735a2* double mutants accumulate iP-type cytokinins but lack tZ, resulting in smaller SAMs, reduced leaf cell numbers, and diminished cambial activity [40], implying that *CYP735A2* is essential for SAM establishment in poplar root suckers.

Conversely, Cluster II contains cytokinin oxidase *CKX5*, the exporter *ABCG14-1*, and the intracellular transporters *PUP4*, *PUP11* and *PUP21-1*, all of which were repressed throughout suckering (Figure 5b), thereby promoting cytokinin accumulation in the root [43]. Cluster III genes (*ABCG14-2*, *LOG1*, and *CKX7*) were up-regulated during late stages of development (D3–D6) (Figure 5b), consistent with roles in meristem regulation via modulation of active cytokinin levels [44,45,46]. LOG enzymes convert the long-distance transport form (tZ riboside, tZR) into active tZ within the SAM, whereas CKXs fine-tune local cytokinin concentrations. Both *LOGs* and *CKXs* function in SAM maturation and maintenance through precise spatial expression. Arabidopsis *LOG4* and *LOG7* are restricted to the L1 layer and CLV3 domain, respectively [22,47], and *CKX5* localizes to the *CLV3-WUS* domain; mutation of *CKX5* enlarges the *WUS* expression zone [22,48,49]. Late activation of *ABCG14-2* may further partition cytokinin between root and newly emerged aerial tissues.

Auxin exhibited a parallel, stage-specific program. The biosynthetic gene *TAR2* was rapidly induced upon suckering initiation, whereas *YUC10* gradually peaked at D6, indicating successive roles in root suckering (Figure 5c). Several *PIN1* and *PIN3* paralogues—key determinants of local auxin gradients [50,51,52]—were differentially expressed, underscoring strict spatiotemporal control of auxin distribution during sucker development. The expression patterns of representative genes involved in the biosynthesis, turnover, and transport of cytokinin and auxin were further confirmed by RT-qPCR (Figure 5d). Collectively, the precisely orchestrated synthesis, transport, and degradation of both cytokinin and auxin ensure normal SAM establishment and sustained organogenesis during poplar root suckering.

Transcript profiling implicated both auxin and cytokinin pathways in sucker formation. To test their functional roles, root segments were treated with either hormone. NAA (0.5 and 1.0 mg L^−1^) reduced sucker number by 47.5% and 59.9% (Figure 6), respectively, confirming that auxin represses sucker development. Conversely, the cytokinin-biosynthesis inhibitor lovastatin suppressed sucker emergence by 60% (Figure 6), demonstrating that cytokinin is required for initiation. Thus, our pharmacological results indicate that auxin suppresses and cytokinin promotes sucker emergence, underscoring that both hormones jointly influence the formation of adventitious shoot during poplar root-suckering.

### 2.4. Reprogramming of Core Meristem Regulators During De Novo SAM Formation in Root Suckering

To elucidate how meristem regulators are reprogrammed during de novo SAM formation on the root, we monitored the expression dynamics of key meristem genes. *WUS*, *STM*, *CLV1*, *BAM1*, *SHR1*, and two *WOX4* paralogues resolved into three clusters (Figure 7a). Cluster I comprises *WUS* and *STM*, which were rapidly up-regulated at D2, which consistents with their roles as stem-cell identity determinants during SAM formation [23,24,25,26]. Cluster II contains *CLV1*, *BAM1*, and *SHR1*, all peaking at D6. In Arabidopsis, the *CLV3*–*CLV1/BAM1* negative-feedback loop limits *WUS* expression to maintain meristem homeostasis [29,30,31], whereas SHR1 directs xylem patterning and cambial activity [53,54,55,56]. These patterns imply that coordinated regulation between stem-cell homeostasis and vascular development during late suckering and post-establishment growth. Cluster III comprises *WOX4-1* and *WOX4-2*, which were induced throughout SAM establishment and organ differentiation (D2–D6). In *Populus*, WOX4 functions as a key cambial regulator [57], and its Arabidopsis and tomato orthologues are essential for vasculature development [58]. Thus, *WOX4-1/2* likely facilitate procambial-strand differentiation and vascular integration between the nascent SAM and the parental root during SAM establishment. The expression patterns of representative genes *STM*, *SHR1* and *WOX4-1* were further confirmed by RT-qPCR (Figure 7b). The specific induction of cambial regulators *WOX4-1/2* during SAM establishment underscores the importance of vascular integration between the nascent shoot and the parental root during root suckering.

## 3. Discussion

We established a rapid, greenhouse-based system that yields abundant suckers within 6 days from 3-year-old white poplar (*Populus davidiana* × *P. bolleana*) roots (Figure 1b), avoiding the months-long induction period of previous system [8]. Daily resolution transcriptomes revealed three successive transcriptional states—primordium initiation (D0–1), SAM establishment (D1–4), and organ differentiation and growth (D4–6) (Figure 3)—and identified auxin–cytokinin–nitrogen crosstalk as the core driver of SAM formation (Figure 4a,b). These findings provide several potential molecular targets for breeding programmes aimed at enhancing or suppressing root-suckering in *Populus*.

### 3.1. Cytological Features and Key Molecular Triggers of Populus Root Suckering

Previous studies in trembling aspen (*Populus tremuloides*) indicate that the initial meristematic activity leading to a sucker primordium occurs in the phellogen and subsequently spreads basipetally through both radial and tangential divisions into the phelloderm and phloem parenchyma [19,59]. In white poplar (*Populus davidiana* × *P. bolleana*), we observed a comparable stage in which 3-day protuberances contained differentiated procambial strands and meristematic cells (Figure 1c). Transcript profiles place *WOX4-1/2*, *SHR1*, and the SAM determinants at this developmental process (Figure 6b). In *Populus*, *WOX4* genes are cambium-specific regulators whose knockdown reduces cambial width and secondary growth [60], therefore their sustained expression here implicates *WOX4-1/2* in procambial-strand differentiation and vasculature development. *SHR1* is likewise expressed in the cambial zone [53], and its Arabidopsis ortholog directs xylem patterning [54,56], suggesting that SHR1 coordinates vasculature development during root suckering. Concomitantly, the SAM determinants *WUS* and *STM* [23,24,25,26] are sharply up-regulated at Day 2 (Figure 7a). This surge coincides with elevated *IPTs* and reduced *CKX5* transcripts (Figure 5b), a combination that raises root cytokinin levels and thereby activates *WUS* expression [27,35,48]. Thus, *WUS* appears to act as a molecular trigger for SAM initiation during poplar root-suckering. A similar role for *WUS* has been reported for SAM formation from root or callus during in vitro regeneration in Arabidopsis [27,28], although the upstream cues may differ. In poplar, the coordinated induction of *WOX4* and *SHR* indicates tight coupling between vasculature development and SAM formation: the procambial strand marked by *WOX4* and *SHR1* differentiates simultaneously with the SAM, ensuring rapid vascular continuity for the nascent shoot. The precise cellular origin of the founding stem cells of the root sucker remains unresolved; single-cell and spatial transcriptomics will therefore be required to trace how specific phellogen derivatives acquire stem-cell identity of SAM and to quantify the contributions of adjacent tissues.

### 3.2. Auxin–Cytokinin–Nitrogen Crosstalk Controls SAM Formation in Populus Root Suckering

In intact plants, auxin is produced predominantly in aerial meristems and transported basipetally, whereas cytokinin is synthesized in the root and exported acropetally via the xylem. Severing the root from the shoot removes the major auxin source and blocks cytokinin efflux, thereby lowering auxin and elevating cytokinin levels within the root—a favourable physiological state for triggering the onset of root suckering [7,18]. Although early physiological studies showed that exogenous auxin suppresses and cytokinin promotes sucker emergence [14], our transcriptome data reveal rapid induction of the auxin biosynthesis-related genes *PAT1*, *TSB1*, and *TSB2* [36,61] during early primordium initiation (Figure 4a,b), indicating that local IAA synthesis from tryptophan is activated; this synthesis is required for cell-cycle re-entry [62]. This implies that root-derived auxin functions differently from the shoot-derived pool during root suckering. In parallel, the auxin-response factor *ARF5* was strongly up-regulated (Figure 4a,b,g); Arabidopsis mutants lacking *AtARF5* exhibit severely impaired de novo shoot regeneration from root explants [35,63], underscoring their likely importance in *Populus* sucker formation.

Strikingly, GO terms related to nitrogen-compound metabolism are enriched during SAM establishment. Enzymes of aspartate and glutamate metabolism (ASP1/2, CTP synthase, GAD, AK2) and glutamine synthetases (GS1-1, GS1-2, GLN2) are continuously up-regulated (Figure 4c–f). Aspartate and glutamate are central intermediates of nitrogen assimilation that modulate the activities of nitrate reductase and glutamine synthetase. Meanwhile, nitrate signaling regulates cytokinin biosynthesis [17,64,65,66]. We therefore hypothesise that those candidate genes related to nitrogen metabolism contributes to meristem formation by modulating cytokinin synthesis. Consistent with this model, mutation of rice *asparagine synthetase 1* (*OsASN1*) lowers aspartate levels and suppresses tiller formation [67]. In human cells, loss-of-function mutations in asparagine synthetase inhibit proliferation [68,69,70], and in rice disruption of *GS1;2* reduces the nitrogen responsiveness of the cytokinin-biosynthetic gene *IPT4* [71,72,73]. In wheat, the cereal-specific NAC transcription factor TaNAC2-5A directly up-regulates glutamine synthetase genes, increasing tiller and spikelet numbers and enhancing grain yield [74]. However, comparative evidence also reveals species specificity of the “nitrate signaling–cytokinin” module. Unlike rice *OsASN1* that directly controls tillering, the sunflower homolog *HaASN1* mainly affects seed germination rather than lateral shoot formation [75]. Similarly, the *Arabidopsis thaliana glutamine synthetase 1;1 1;2* (*atgs1;1 atgs1;2*) double mutant exhibits reduced lateral roots under low-nitrogen conditions, with no obvious SAM defects [76]. These observations suggest that the functional link between nitrogen assimilation and cytokinin-driven bud outgrowth may be context- or taxon-dependent. Whether the strong transcriptional induction of *GS1-2* and *ASN1* during poplar root suckering reflects a unique, woody-specific regulatory circuit therefore awaits validation by CRISPR/Cas9-mediated gene editing in poplar. Therefore, functional validation of the candidate genes involved in auxin and cytokinin biosynthesis and signaling, as well as nitrogen metabolism, will be required to confirm their contributions to root suckering and to inform future molecular breeding strategies.

## 4. Materials and Methods

### 4.1. Plant Material and Induction of Root Suckers

Roots of *Populus davidiana* × *P. bolleana* were collected during autumn dormancy following cessation of growth. Segments (~10 cm in length) were inserted upright into pre-moistened substrate (imported peat:compost:vermiculite = 2:2:1, *v*/*v*), with ~5–7 cm buried and the remainder left aerial. Plantings were maintained under dark conditions in a greenhouse at 25 °C and 70 ± 5% relative humidity. Visible root suckers emerged from the aerial portion of the segments within 7 d.

### 4.2. Semi-Thin Sectioning and Microscopy

Root dermal tissues bearing white, dome-shaped protuberances (~1 cm long) were excised distal to the vascular cambium and fixed in FAA (70% ethanol, 37% formaldehyde, glacial acetic acid; 90:5:5 *v*/*v*). After vacuum infiltration for 30 min, samples were transferred to fresh FAA and stored at 4 °C for 48 h. Tissues were dehydrated through an ethanol series (1 h each in 70%, 85%, 95%, 100%, and 100% ethanol), infiltrated stepwise with LR White resin diluted in ethanol (25%, 33%, 50%, 75% resin, 1 h each), then incubated three times in 100% LR White resin (≥12 h each). Samples were placed in gelatin capsules filled with LR White resin and polymerized at 37 °C for 1 week. Blocks were trimmed, and 3 µm-thick sections were cut on a Leica Ultracut R ultramicrotome (Leica Microsystems GmbH, Wetzlar, Germany), floated onto adhesive slides with sterile water, and dried at 60 °C. Sections were stained with 0.1% toluidine blue O (TBO) and examined by bright-field microscopy (Olympus BX53, Olympus Corporation, Tokyo, Japan).

### 4.3. RNA Extraction, cDNA Library Construction and High-Throughput Sequencing

Root segments were collected in the field (collection day designated Day 0), immediately transferred to the greenhouse, and grown in darkness as described above. Tissues were harvested daily at 11:00 a.m. from Day 0 to Day 6. Before visible bud emergence, root dermal tissue distal to the vascular cambium was excised; once protuberances were macroscopically evident, the entire sucker primordium or adventitious bud was dissected distal to the cambium. For each time point, nine samples from nine individual plants were pooled into three biological replicates (three samples per replicate), snap-frozen in liquid nitrogen, and stored at −80 °C.

Total RNA was extracted using the RNAprep Pure Plant Kit (Tiangen Biotech Co., Ltd., Beijing, China; Cat. DP441) in accordance with the manufacturer’s instructions. RNA integrity was assessed with an Agilent 2100 Bioanalyzer (Agilent Technologies, Santa Clara, CA, USA), and only samples with an RNA Integrity Number (RIN) ≥ 8.5 were used for subsequent library preparation. Strand-specific cDNA libraries were constructed following poly (A) selection using the NEBNext^®^ Ultra™ RNA Library Prep Kit (New England Biolabs, Ipswich, MA, USA). Sequencing was performed on DNBSEQ-T7 platform (150 bp paired-end reads) by IGENEBOOK Biotechnology Co., Ltd. (Wuhan, China). Raw sequencing reads were quality-checked with FastQC v0.11.5, and high-quality clean reads were obtained for downstream analysis.

### 4.4. Bioinformatic Analysis

Clean reads were aligned to the *Populus trichocarpa* v4.1 reference genome using HISAT2 v2.0.1-beta. Gene-level read counts were quantified with featureCounts v1.6.0. Expression values were normalised to FPKM (Fragments Per Kilobase of exon per Million mapped reads) [77]. Principal component analysis (PCA) and Pearson correlation coefficients were calculated to assess sample relationships. Differentially expressed genes (DEGs) were identified with edgeR v3.38.4 [78], using a false-discovery rate (FDR) < 0.05 and |log2 fold-change| ≥ 1.

### 4.5. Weighted Gene Co-Expression Network Analysis (WGCNA)

WGCNA was performed using the WGCNA plug-in in TBtools v1.098 [79]. A signed network was built using the automatic soft-thresholding power that maximised scale-free topology (R^2^ > 0.8). Modules were delimited by dynamic tree cut (minModuleSize = 100). Eigengenes of resulting modules were correlated with developmental stages to quantify module–trait relationships. Modules significantly associated with specific stages (|r| ≥ 0.44, *p* < 0.05) were subjected to Gene Ontology (GO) and Kyoto Encyclopedia of Genes and Genomes (KEGG) pathway enrichment to identify biological processes and hub genes potentially involved in root suckering in *Populus*.

### 4.6. Reverse Transcription Quantitative Polymerase Chain Reaction (RT-qPCR)

Genomic DNA was removed from total RNA with DNase I, and first-strand cDNA was synthesized using a commercial reverse-transcription kit (Catalog No: AG11705, Accurate Biology, Hunan, China). Gene-specific primers spanning conserved domains were designed with national center for biotechnology information. Reverse transcription quantitative polymerase chain reaction (RT-qPCR) was performed on a Bio-Rad CFX system using SYBR Green chemistry (Bio-Rad Laboratories, Hercules, CA, USA); amplicon specificity was verified by melting-curve analysis.

### 4.7. Sucker Induction Under Hormone Treatments

Root segments were incubated in continuous darkness at 25 °C for 6 days. Each segment was immersed in one of the following sterile solutions: 0.5 or 1.0 mg mL^−1^ 1-naphthaleneacetic acid (NAA) or 5.0 mg mL^−1^ lovastatin (cytokinin-biosynthesis inhibitor). A minimum of 13 biological replicates was used per treatment. At the end of the incubation period, emerging root suckers were counted and photographed.

## 5. Conclusions

By coupling a six-day root-suckering induction system established here with daily resolution transcriptomes, we provide the first integrated atlas of *Populus* root suckering. Three sequential stages—primordium initiation, SAM establishment, and organ differentiation—are orchestrated by auxin–cytokinin–nitrogen crosstalk that sequentially activates *WUS*/*STM*, *WOX4-1*/*2*, and *CLV1*/*BAM1–SHR1* networks. This spatiotemporal programme simultaneously drives de novo meristem formation and vascular integration between the nascent shoot and the parental root. The findings identify targets for molecular breeding to enhance clonal propagation efficiency or curb unwanted sucker emergence in *Populus* plantations, accelerating the sustainable deployment of genetically improved *Populus* clones.

## Figures and Tables

**Figure 1 ijms-26-12172-f001:**
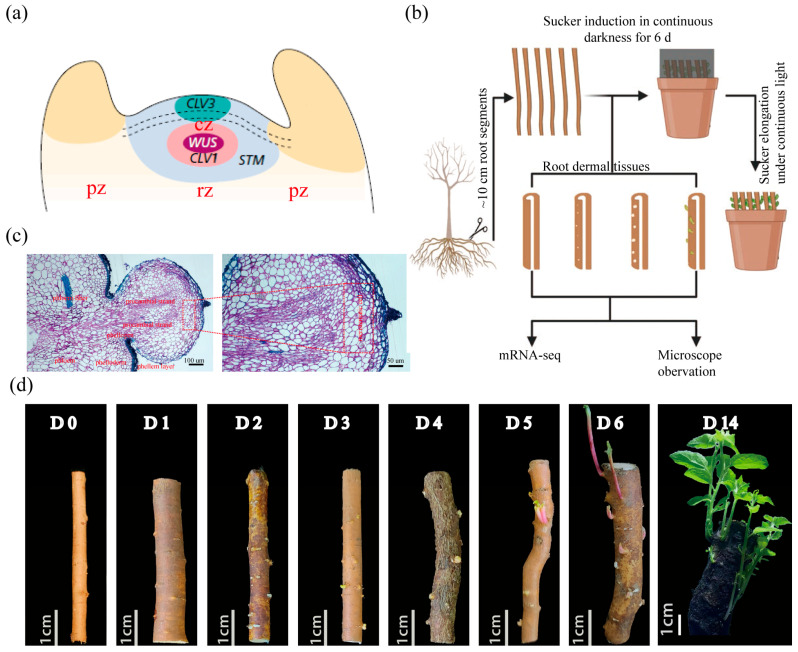
Rapid induction and temporal dynamics of root suckering in *Populus davidiana × P. bolleana*. (**a**) Canonical shoot apical meristem (SAM) schematic [29]. *WUS* and *STM* maintain central-zone (CZ) stem cells; the *CLV3–CLV1* loop restrains *WUS* to maintain SAM homeostasis. CZ, central zone; RZ, rib zone; PZ, peripheral zone. (**b**) Experimental design. ~10 cm root segments from 3-year-old trees were incubated in darkness; suckers emerged from the aerial portion within 6 d. Daily samples distal to the vascular cambium were split for RNA-seq and histology. (**c**) Semi-thin longitudinal section of day 3 (D3) samples showing a root-borne primordium enveloped by phellem. Two procambial strands link the primordium to the root vascular cambium; the apical meristematic cluster will form the SAM. Scale bars: 100 µm (**left**), 50 µm (**right**). (**d**) Macroscopic progression of root suckering: white ellipsoidal protuberances at D3, shoot buds with juvenile leaves at D5–6, and elongating shoots by D14. Scale bar: 1 cm.

**Figure 2 ijms-26-12172-f002:**
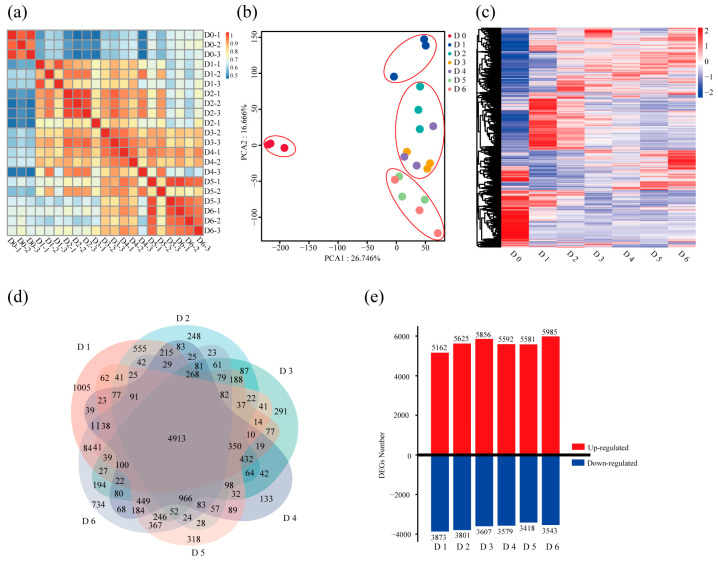
Transcriptomic dynamics during root suckering. (**a**) Pearson correlation coefficients (0.945–1.000) between biological replicates indicate high reproducibility. (**b**) Principal component analysis reveals four distinct transcriptional states that group the samples into D0, D1, D2–4, and D5–6, defining three developmental stages: (i) primordium initiation (D0–1), (ii) SAM establishment (D1–4), and (iii) organ differentiation and growth (D4–6). (**c**) Expression abundance of genes detected in all samples. (**d**) Venn diagram of differentially expressed genes (DEGs) from each pairwise comparison (D1–D6 vs. D0), showing 4913 common DEGs. (**e**) Bar plot of up-regulated (red) and down-regulated (blue) DEGs at each time point, illustrating extensive transcriptional reprogramming during development.

**Figure 3 ijms-26-12172-f003:**
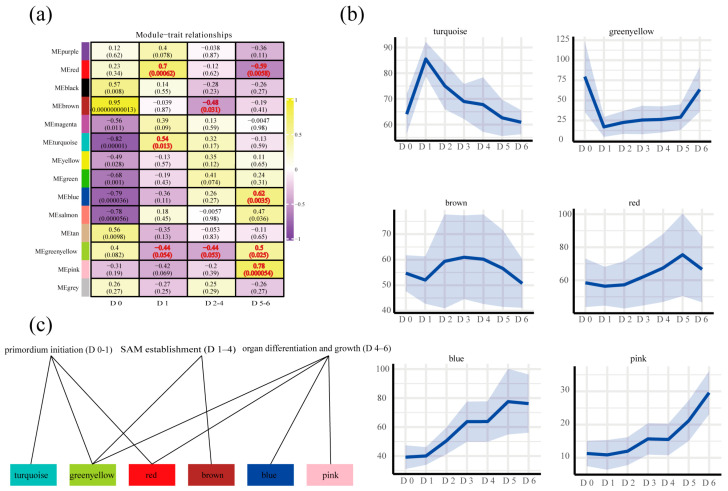
Module–trait relationships revealed by weighted gene co-expression network analysis (WGCNA) during root suckering. (**a**) Heatmap of Pearson correlations between each module and the suckering stages. Correlation coefficients (r, upper value; range −1 [purple] to +1 [yellow]) and associated *p*-values (in parentheses) are shown; red font denotes |r| ≥ 0.44 and *p* < 0.05. The grey module contains unassigned genes. (**b**) Eigengene expression profiles of the six stage-associated modules. This shadow represents the expression trends of numerous genes under this module. (**c**) Summary of module assignments to specific developmental stages.

**Figure 4 ijms-26-12172-f004:**
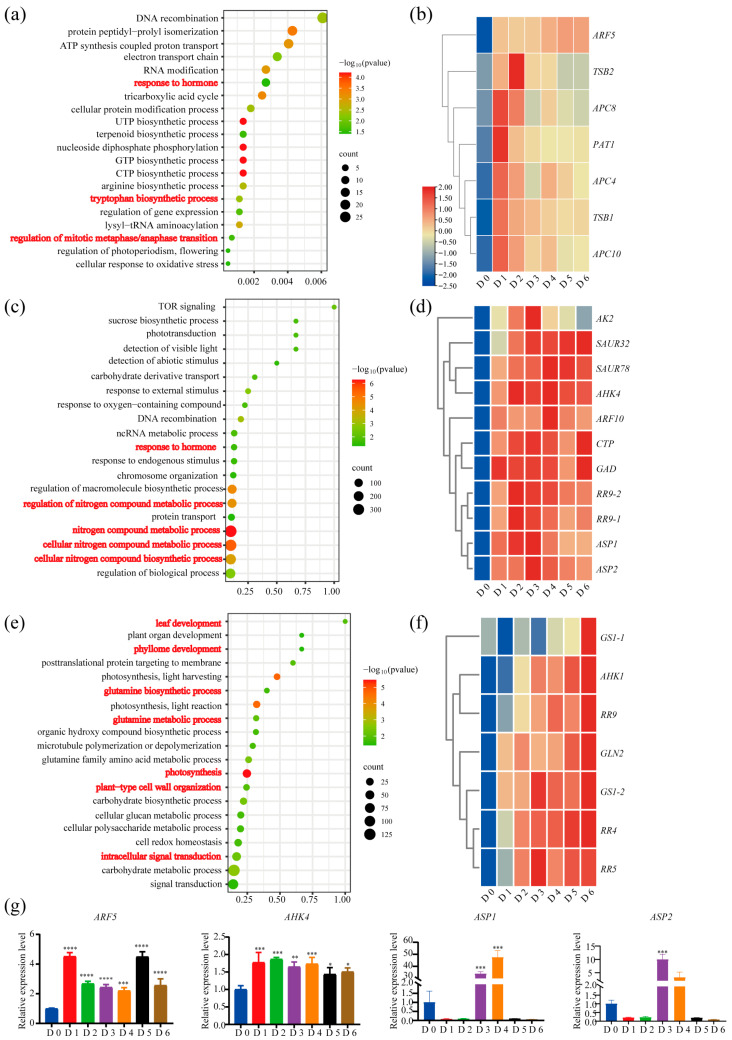
Gene Ontology (GO) enrichment and expression profiles of key genes across the three developmental stages of root suckering. (**a**,**c**,**e**) Bubble plots showing significantly enriched GO terms during primordium initiation (**a**), SAM establishment (**c**), and organ differentiation and growth (**e**). (**b**,**d**,**f**) Heatmaps of key genes associated with primordium initiation (**b**), SAM establishment (**d**), and organ differentiation and growth (**f**). (**g**) Reverse transcription quantitative PCR (RT-qPCR) validation of selected gene expression patterns. Significance was assessed by one-way ANOVA (*, *p* < 0.05; **, *p* < 0.01; ***, *p* < 0.001; **** *p* < 0.0001). Error bars represent SD of three biological replicates.

**Figure 5 ijms-26-12172-f005:**
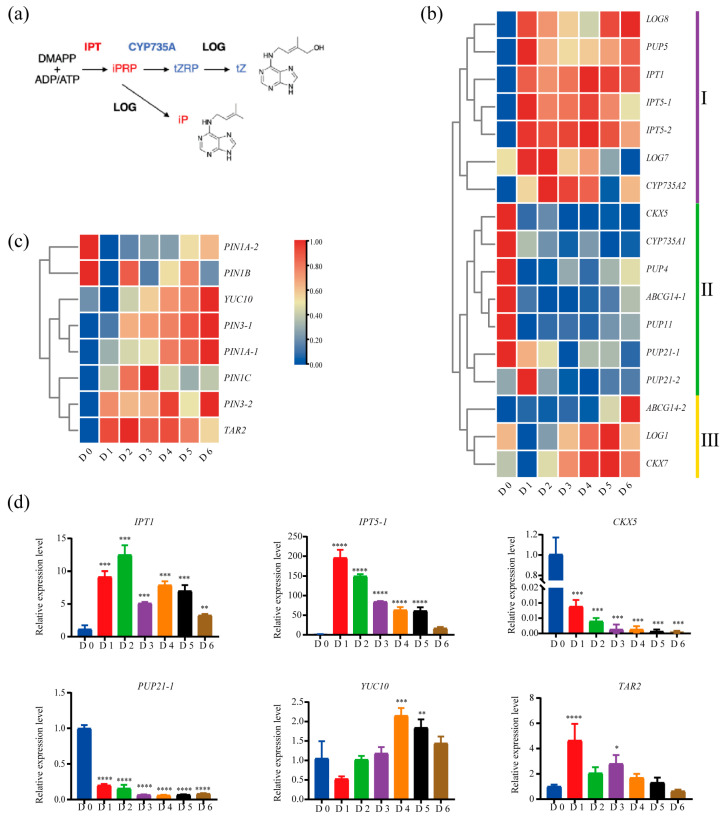
Expression dynamics of representative genes related to auxin and cytokinin pathways during root suckering. (**a**) Simplified schematic of the cytokinin biosynthesis pathway [41]. (**b**,**c**) Heatmaps of cytokinin- (**b**) and auxin-related genes (**c**) showing expression dynamics during root suckering. (**d**) RT-qPCR validation of selected gene expression patterns. Significance was assessed by one-way ANOVA (*, *p* < 0.05; **, *p* < 0.01; ***, *p* < 0.001; **** *p* < 0.0001). Error bars represent SD of three biological replicates.

**Figure 6 ijms-26-12172-f006:**
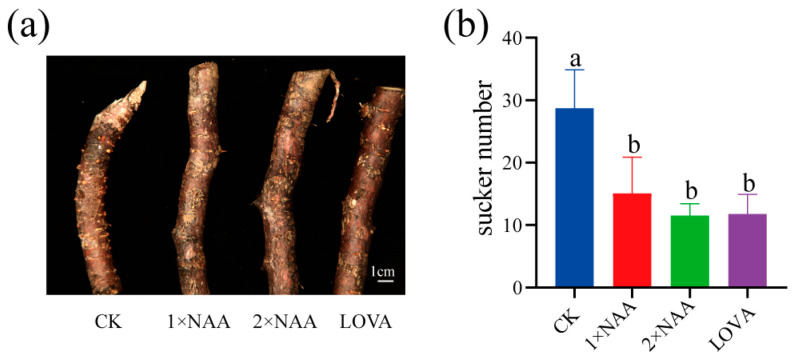
Auxin and cytokinin affects sucker initiation of poplar. (**a**) Representative images of root segments treated with 1× or 2 × NAA (0.5 or 1.0 mg L^−1^) and 5.0 mg L^−1^ LOVA (cytokinin-biosynthesis inhibitor). (**b**) Quantification of the root-sucker number data in (**a**). The data are presented as means ± SE (*n* ≥ 13). Different letters indicate statistically significant differences across genotypes, while the same letter indicates no significant difference according to one-way ANOVA Duncan’s (D) test (*p* < 0.05).

**Figure 7 ijms-26-12172-f007:**
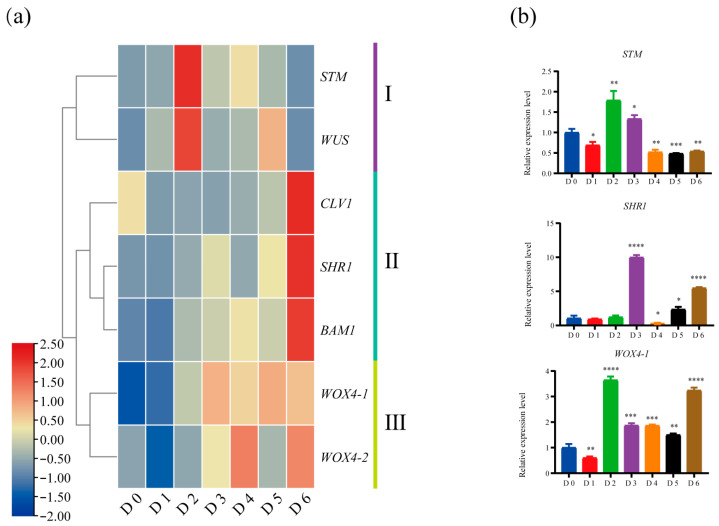
Reprogramming of core meristem regulators during de novo SAM formation in root suckering. (**a**) Expression dynamics of key meristem regulatory genes during root suckering. (**b**) RT-qPCR validation of selected gene expression patterns. Significance was assessed by one-way ANOVA (*, *p* < 0.05; **, *p* < 0.01; ***, *p* < 0.001; **** *p* < 0.0001). Error bars represent SD of three biological replicates.

## Data Availability

The data presented in this study are openly available in National Center for Biotechnology Information Sequence Read Archive, reference number PRJNA1332294.

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
