# Peer review of "Coordinated Auxin–Cytokinin–Nitrogen Signaling Orchestrates Root Suckering in Populus"

_ijms, 2025, doi:10.3390/ijms262412172_

Round 1

Reviewer 1 Report

Comments and Suggestions for Authors

Root suckering is a critical pathway for the clonal propagation of aspens (section Populus) and plays a vital role in their commercial application. This study employed transcriptome sequencing to investigate the molecular mechanisms underlying root suckering in aspens, which provides a foundational resource for mechanistic understanding. The writing is clear, the figures are elegant, and the results are easy to understand. There are still some questions. 1. The first sentence of Abstract:” Populus spp. rely on root suckers for clonal propagation”. The poplar genus is divided into five main groups, and not all of them reproduce via root suckers. The white poplar group (aspen) mainly propagates through root suckering, but the black poplar group is reproduced using cuttings. Hence, the original statement is inaccurate. 2. The abbreviations "SAM" and "STM" are used in the abstract without first being defined. Please spell out the full term upon first use, followed by the abbreviation in parentheses. 3. Figure 1c: How can you confirm that the cells circled in the box represent the shoot apical meristematic cells? 4. The statement, "Thus, a precisely balanced auxin–cytokinin network controls the timing and extent of shoot apical meristem formation during poplar root-suckering," appears to be an overinterpretation. The data demonstrates the importance of both hormones, but does not fully elucidate the precise, dynamic balance implied by this conclusion. It is recommended to temper this language to more accurately reflect the experimental findings.

Author Response

Comment 1: The first sentence of Abstract:“Populus spp. rely on root suckers for clonal propagation”. The poplar genus is divided into five main groups, and not all of them reproduce via root suckers. The white poplar group (aspen) mainly propagates through root suckering, but the black poplar group is reproduced using cuttings. Hence, the original statement is inaccurate.

Response: We sincerely thank the reviewer for this excellent and precise correction. We completely agree with the reviewer that our original statement was overly broad and inaccurate. Not all species within the Populus genus exhibit strong root suckering behavior. As correctly pointed out, this trait is predominant in the white poplar / Sect. Populus group, such as aspen. We have therefore revised lines 13–14 to accurately reflect this. The revised text now specifies the relevant groups. The change has been highlighted in the revised manuscript.

Comment 2: The abbreviations "SAM" and "STM" are used in the abstract without first being defined. Please spell out the full term upon first use, followed by the abbreviation in parentheses.

Response: We thank the reviewer for pointing out this oversight. We have now corrected the text in line 18-19, 28-29 and 35-36. The full terms have been spelled out upon first use, followed by their abbreviations in parentheses, as suggested. The changes have been highlighted in the revised manuscript.

Comment 3: Figure 1c: How can you confirm that the cells circled in the box represent the shoot apical meristematic cells?

Response: We appreciate the reviewer's question regarding the identification of the SAM. Our identification is based on well-established anatomical criteria: the SAM initials are located at the distal tip of the primordium, directly above the procambial strands— the canonical position for SAM.

We have revised lines 118–121 accordingly; the changes are highlighted in the revised manuscript.

Comment 4: The statement, "Thus, a precisely balanced auxin–cytokinin network controls the timing and extent of shoot apical meristem formation during poplar root-suckering," appears to be an overinterpretation. The data demonstrates the importance of both hormones, but does not fully elucidate the precise, dynamic balance implied by this conclusion. It is recommended to temper this language to more accurately reflect the experimental findings.

Response: We sincerely thank the reviewer for this critical and insightful comment. We agree that the phrase "precisely balanced" may overstate the mechanistic resolution of our current data, as our study does not define the exact stoichiometry or real-time dynamics of the hormone interaction. We have therefore toned down our language to more accurately reflect our experimental findings in line 295-298.

Reviewer 2 Report

Comments and Suggestions for Authors

Dear Authors,

I read with great interest the manuscript by Hongying Pang et al. entitled "Coordinated auxin–cytokinin–nitrogen signaling orchestrates root suckering in Populus."
This article presents a study on the molecular mechanisms of root sucker formation in poplars. The work is characterized by a well-designed experiment, the use of modern transcriptome analysis methods, and testing of the functional roles of hormones.

Some comments and suggestions:

The work extensively references numerous genes and their functions, but often lacks a discussion of possible alternative interpretations or limitations of existing knowledge. Adding a critical discussion section would enhance the depth of the study.

While the introduction references previous studies, the Discussion section would benefit from a more direct comparison of the obtained results with similar processes in other plant species or under different types of regeneration. This would help contextualize the significance of the identified pathways in poplar.

Refining these points will help authors significantly improve the quality of the submitted work.

Author Response

Comment 1: The work extensively references numerous genes and their functions, but often lacks a discussion of possible alternative interpretations or limitations of existing knowledge. Adding a critical discussion section would enhance the depth of the study.

Response: We sincerely thank the reviewer for this excellent suggestion to improve the scholarly rigor of our manuscript. In response, we have significantly expanded the critical discussion of alternative interpretations and limitations in line 401-411.

Comment 2: While the introduction references previous studies, the Discussion section would benefit from a more direct comparison of the obtained results with similar processes in other plant species or under different types of regeneration. This would help contextualize the significance of the identified pathways in poplar.

Response: Thank you for this valuable suggestion. In the revised Discussion we have added a direct cross-species comparison to better position our findings in poplar within the broader context of plant regeneration in line 357-366.

Round 2

Reviewer 2 Report

Comments and Suggestions for Authors

Thank you, I think that the article can be accepted in its present form.